# Gas within the Intervertebral Disc Does Not Rule Out Spinal Infection—A Case Series of 135 Patients with Spontaneous Spondylodiscitis

**DOI:** 10.3390/diagnostics12051089

**Published:** 2022-04-27

**Authors:** Friederike Schömig, Zhao Li, Luis Becker, Tu-Lan Vu-Han, Matthias Pumberger, Torsten Diekhoff

**Affiliations:** 1Center for Musculoskeletal Surgery, Charité—University Medicine Berlin, Charitéplatz 1, 10117 Berlin, Germany; zhao.li@charite.de (Z.L.); luis-alexander.becker@charite.de (L.B.); tu-lan.vu-han@charite.de (T.-L.V.-H.); matthias.pumberger@charite.de (M.P.); 2Department of Radiology, Charité—University Medicine Berlin, Charitéplatz 1, 10117 Berlin, Germany; torsten.diekhoff@charite.de

**Keywords:** spine surgery, infection, spondylodiscitis, diagnosis, vacuum phenomenon

## Abstract

Gas in the intervertebral disc is mainly associated with degenerative disc diseases and experts generally assume that it is unlikely in spinal infection. However, large-scale studies supporting this notion are lacking, which is why our study’s aim was to analyze the prevalence of and factors associated with the occurrence of gas in patients with spontaneous spondylodiscitis. Patients presenting with spontaneous spondylodiscitis from 2006 to 2020 were included retrospectively. Exclusion criteria were previous interventions in the same spinal segment and missing imaging data. Clinical data were retrieved from electronic medical reports. Computed tomography (CT) scans were evaluated for the presence of intervertebral gas. Causative pathogens were identified from CT-guided biopsy, open biopsy, intraoperative tissue samples, and/or blood cultures. 135 patients with a mean age of 66.0 ± 13.7 years were included. In 93 patients (68.9%), a causative pathogen was found. Intervertebral gas was found in 31 patients (23.0%) in total and in 19 patients (20.4%) with positive microbiology. Patients with gas presented with significantly higher body temperatures (37.2 ± 1.1 vs. 36.8 ± 0.7 °C, *p* = 0.044) and CRP levels (134.2 ± 127.1 vs. 89.8 ± 97.3 mg/L, *p* = 0.040) on admission. As a considerable number of patients with spondylodiscitis showed intervertebral gas formation, the detection of intervertebral gas is not suited to ruling out spondylodiscitis but must be interpreted in the context of other imaging and clinical findings, especially in elderly patients.

## 1. Introduction

Intervertebral vacuum phenomenon is defined as a collection of gas within the intervertebral disc and was first described by Knutsson as a pathognomonic sign of disc degeneration in 1942 [1,2]. Its prevalence in the literature ranges from 2–20% with a higher frequency in the aging population [3,4,5]. The pathogenesis of intervertebral gas is still under debate. However, several studies suggest that it occurs as the result of a dynamic process involving the balance between the disc’s liquid and gaseous components [3,6,7]. It may develop either acutely when a rapid increase in volume is caused by external forces, or as a chronic process in increasingly permeable tissue, which allows a progressive penetration of gas and/or liquid due to a reduction in volume [8]. The resulting intradiscal gas collections contain 90–92% nitrogen combined with oxygen, carbon dioxide and other traces of gases [9,10].

Even though magnetic resonance imaging (MRI) is the standard of reference for most spinal pathologies, it has a low sensitivity in detecting gas [3]. While radiography may detect gas, computed tomography (CT) has the highest sensitivity compared with other imaging modalities and, therefore, is accepted as the standard of reference [7].

To date, gas within the intervertebral disc is often assumed to be associated with degenerative disc disease [11]. However, previous studies have shown that intervertebral gas may also occur in other pathologies such as traumatic injuries of the spine [12,13]. In spinal infections, however, only isolated case studies show gas formations in the intervertebral disc caused predominantly by gas-forming microorganisms [2,14]. Thus, it has been proposed that the identification of gas within the disc is an important imaging sign to rule out spondylodiscitis or, at least, make an infection highly unlikely [8,15]. 

Despite a rising incidence of spondylodiscitis, its diagnosis and thereby treatment is often delayed by several months due to a lack of specific symptoms of spinal infection [16,17]. While the incidence of spinal infections has been steadily increasing and is still associated with high morbidity and mortality, larger studies investigating gas in spondylodiscitis are lacking [18]. As effective treatment of spinal infections depends on early diagnosis and pathogen identification, a better understanding of associated radiological features is essential for early identification and thereby treatment. Thus, our study’s aim was to analyze the prevalence and factors correlated with the occurrence of gas in CT scans of patients with confirmed spontaneous spondylodiscitis. 

## 2. Materials and Methods

The study was approved by the institutional ethics committee (EA1/019/21). Patients presenting with spontaneous spondylodiscitis between January 2006 and December 2020 were identified using information from discharge letters and diagnostic or procedural codes and were included retrospectively. Previous interventions in the same spinal segment or incomplete imaging data led to patient exclusion. 

Clinical data including demographics, clinical characteristics including body-mass-index (BMI), body temperature and pain, comorbidities, Charlson Comorbidity Index (CCI) scores, laboratory parameters (C-reactive protein, CRP; white blood cell count, WBC; hemoglobin, Hb), and treatment details were retrieved from electronic medical reports and patient charts. Pain was assessed by visual analog scale (VAS) scores. Pathogens were identified from CT-guided biopsy, open biopsy, intraoperative tissue sampling, and/or blood cultures. 

According to the Infectious Diseases Society of America guidelines, spondylodiscitis was defined as a combination of radiological changes of the intervertebral disc and adjacent vertebrae, as well as possible abscess formation, in MRI and/or CT scans and clinical findings such as elevated CRP or WBC levels, back or neck pain, and fever [19]. Spondylodiscitis of the cervicothoracic junction (C7/Th1) was defined as cervical, the thoracolumbar junction (Th12/L1) as thoracic, and the lumbosacral junction (L5/S1) as lumbar spondylodiscitis. 

Patients were “gas-positive” if gaseous radiolucency was identified at the same level as spondylodiscitis in CT images. CT and MR imaging was evaluated by an orthopedic surgery resident and a research fellow specializing in musculoskeletal imaging. Disagreement was solved by the consensus of an orthopedic surgeon with eleven years of experience and an orthopedic surgery resident with three years of experience. The presence of gas was confirmed by a radiologist specializing in musculoskeletal disorders with eleven years of experience. Patients were then grouped into gas-positive and gas-negative groups.

For statistical analysis, descriptive summaries were analyzed using Pearson Chi-squared or Fisher’s exact test for nominal, Mann–Whitney U test for ordinal, or Student t-test for continuous variables. Diagnostic accuracies of microbiological tissue sampling methods were analyzed by cross tabulation with intraoperative tissue sampling as the diagnostic standard. Statistical analyses were performed using SPSS version 27 (SPSS Inc., Chicago, Illinois). For all tests, a *p*-value of < 0.05 was considered significant.

## 3. Results

A total of 247 patients with spondylodiscitis were identified. Fifty-four patients were excluded due to previous interventions at the same segment, and 58 due to missing CT imaging, yielding a total of 135 included patients (Figure 1).

Sixty-one patients (45.2%) were female. The mean patient age was 65.9 ± 13.6 years, with a mean BMI of 26.7 ± 6.1 kg/m^2^. Seven patients (5.2%) had spondylodiscitis of the cervical spine, 37 patients (27.4%) of the thoracic, and 81 patients (60.0%) of the lumbar spine. In ten cases (7.4%), spondylodiscitis was disseminated. In all included patients, diagnosis of spondylodiscitis was based on a combination of radiological changes of the intervertebral disc and adjacent vertebrae, as well as possible abscess formation in MRI and/or CT scans and clinical findings such as elevated CRP or WBC levels, back or neck pain, and fever. In 93 patients (68.9%) a causative pathogen was found. Thirty-five patients (25.9%) were treated conservatively, in 17 (48.6%) of which, a causative pathogen was found by CT-guided or open biopsy. One hundred patients (74.1%) received surgical treatment. In 76 (76.0%) of these patients, a causative pathogen was found. Methods performed for pathogen identification are shown in Table 1.

Gas was found in 31 patients (23.0%) in total and in 19 patients (20.4%) with positive microbiology. CT scans showing intervertebral gas in two patients with spontaneous spondylodiscitis are presented in Figure 2.

In 105 patients (77.8%), additional MR imaging was performed. For the presence of an epidural abscess (54.2% vs. 65.4%, *p* = 0.316), a paravertebral abscess (58.3% vs. 50.6%, *p* = 0.506), or fluid within the disc (83.3% vs. 87.7%, *p* = 0.584), no significant differences were found between the gas-positive and gas-negative groups, respectively.

Patient characteristics of gas-positive and gas-negative patients are presented in Table 2.

Patients with gas were significantly older (71.9 ± 10.8 vs. 64.0 ± 13.9 years, *p* = 0.004) and had significantly higher CCI scores (3.9 ± 2.9 vs. 2.7 ± 2.8, *p* = 0.044) with a significantly higher frequency of cancer (48.4% vs. 26.9%, *p* = 0.024). Furthermore, patients with intervertebral gas presented with a significantly higher initial body temperature (37.2 ± 1.1 vs. 36.8 ± 0.7 °C, *p* = 0.044) and had significantly higher CRP levels (134.2 ± 127.1 vs. 89.8 ± 97.3 mg/L, *p* = 0.040).

A total of 47 patients underwent CT-guided biopsy with 25 (53.2%) positive results. An open biopsy was performed in 45 patients with 31 (68.9%) positive results. Intraoperative tissue samples were analyzed from 88 patients with 57 (64.8%) positive results. Cross tabulation analysis with intraoperative tissue sampling as the diagnostic standard yielded a sensitivity of 0.92 and specificity of 0.77 for open biopsy, and a sensitivity of 0.80 and specificity of 0.33 for CT-guided biopsy. 

Pathogen spectra are shown in Table 3. In 19 (61.3%) gas-positive patients and 74 (71.2%) gas-negative patients at least one pathogen was detected. Polymicrobial infections were found in six (31.6%) gas-positive and four (5.4%) gas-negative patients. Enterococci were found significantly more frequently in patients with intervertebral gas (*p* = 0.026). Other gram-positive pathogens included *Aerococcus urinae* and *Trueperella bernardiae*. Other gram-negative pathogens included *Enterobacter aerogenes*, *Proteus mirabilis*, *Bacteroides capillosus*, *Klebsiella pneumoniae*, *Fusobacterium nucleatum*, *Hafnia alvei*, and *Raoultella planticola*.

## 4. Discussion

To our knowledge, this is the first large-scale study investigating the frequency of intervertebral gas occurring in CT images of patients with spontaneous spondylodiscitis. While, to date, it has been assumed that the presence of gas in the intervertebral disc is a stark argument against the presence of a spinal infection, our results show unexpectedly high rates of intervertebral gas with frequencies of 23.0% in all patients with spontaneous spondylodiscitis and 20.4% in microbiology-positive patients [8,15]. Thus, we postulate that, in contrast to previous assumptions, spinal gas formation and infection may, in fact, occur simultaneously and that the presence of intervertebral gas does not rule out the possibility of spondylodiscitis. Furthermore, intervertebral gas in patients with spontaneous spondylodiscitis is associated with more severe symptoms on admission.

Our results show that patients with spondylodiscitis may not only show intervertebral gas but present with significantly higher initial body temperatures (37.2 ± 1.1 vs. 36.8 ± 0.7 °C, *p* = 0.044) and significantly higher initial CRP levels (134.2 ± 127.1 vs. 89.8 ± 97.3 mg/L, *p* = 0.040) compared with the gas-negative group. Therefore, spondylodiscitis needs to be considered as a possible diagnosis especially in cases where symptoms are suggestive of spondylodiscitis, in combination with intervertebral radiolucency on CT imaging. Both elevated body temperature and CRP levels may also be indicative of a more advanced disease stage in patients with pre-existing degenerative changes masking the initial symptoms of spondylodiscitis. 

Due to low specificity of infectious parameters, diagnosis needs to be based on a combination of parameters including radiological changes and clinical findings, as published by the Infectious Diseases Society of America [19]. Definitive diagnosis still depends on microbiological examination of infected tissue [20]. Here, we found open biopsy to be more accurate compared with CT-guided biopsy, with intraoperative tissue sampling as the diagnostic standard. However, only 16 patients had both a CT-guided biopsy and intraoperative tissue sampling and only 37 patients had both an open biopsy and intraoperative tissue sampling, which is why these results need to be interpretated cautiously. While we did find enterococci with a significantly higher frequency in gas-positive patients (*p* = 0.026), these results need to be discussed with caution as they are based on a low number of patients, with only three patients in the gas-positive and one patient in the gas-negative group. Thus, the statistical power of these analyses is rather low which is why no final conclusion regarding an association between causative pathogens and the occurrence of intervertebral gas can be drawn from our results.

We did not find significant changes in treatment choices made in gas-positive and gas-negative patients in terms of whether conservative or surgical treatment was performed (*p* = 0.986). While this suggests that the presence of intervertebral gas has limited clinical implications, it is important to note that, in spondylodiscitis, there is still a lack of widely accepted guidelines for surgical treatment decisions. As in degenerative disc diseases, intervertebral gas is regarded as a sign of instability, and long-term follow-up data of patients treated for spondylodiscitis presenting with intervertebral gas need to be analyzed to make a final statement on its consequences for treatment strategies [11,21].

In a previous study, Feng et al. investigated the frequency of intervertebral gas in plain radiographs of patients with pyogenic or tuberculous spinal infections. In the 317 included patients, they found intervertebral gas in only one patient with tuberculous spondylitis [13]. However, as proven by multiple previous studies, CT has a higher sensitivity for the differentiation of gas than plain radiographs, partly because CT scans are obtained in the supine position [6,22]. Thus, the sole analysis of plain radiographs may explain the low frequency of intervertebral gas in their study.

As in the current literature, there are only rare reports of intradiscal or intraosseous gas formations accompanying spinal infections, it has been widely assumed that gas is unlikely in spinal infection and restricted to a small number of pathogens [15]. Most of the existing literature on radiolucent collections seen in infections are associated with gas-producing microorganisms [2,14,22]. However, this literature is limited to only a few case reports, suggesting that the phenomenon has not been well studied. 

Intervertebral gas has mostly been studied in the context of disc degeneration, postulating multiple theories regarding its pathogenesis. It is presumed that intradiscal clefts are formed due to aging or chronic injury, with impaired disc nutrition causing a reduction in disc volume [23]. This reduction in volume, in turn, leads to a diffusion of gas into the disc from neighboring tissues. In case of disc compression, gas and/or fluid may be re-injected into neighboring tissues, which then results in a so-called pumping phenomenon [24]. In acute intervertebral gas formations, on the other hand, external forces cause a rapid volume increase due to a crack in the outer fibers of the annulus fibrosus, which, however, does not cause the entering of gas or fluid [8]. 

As a sign of disc degeneration, studies have shown an increasing prevalence of intervertebral gas in the aging population [22]. This is in line with our results, in which patients with intervertebral gas were significantly older than patients without intervertebral gas (71.9 ± 10.8 vs. 64.0 ± 13.9 years, *p* = 0.004). In line with the existing theories on the pathogenesis of intervertebral gas formation, we assume that this difference in age may be a sign of pre-existing degenerative changes in the affected discs, which increases the probability of intervertebral gas development. Due to an already injured disc with possibly impaired nutrition, further mechanical stresses caused by infectious spinal alterations may lead to intradiscal clefts. The fact that the gas-positive group also showed significantly higher CCI scores (3.9 ± 2.9 vs. 2.7 ± 2.8, *p* = 0.044) and was significantly more frequently diagnosed with cancer (48.4% vs. 26.9%, *p* = 0.024) may also be explained by the significantly higher mean age of these patients.

This work does not have the claim to distinguish between gas collections primarily caused by degenerative disc disease and gas collections caused by the infection itself. However, as we show intervertebral gas occurring in up to 23.0% of patients presenting with spontaneous spondylodiscitis, this phenomenon is not suitable to rule out spinal infection as a possible diagnosis. This is especially important as clinical signs of spondylodiscitis are unspecific and diagnosis, therefore, is often delayed by up to several months [16,17]. Furthermore, previous studies have shown both higher CCI scores and high CRP levels at diagnosis to be significantly correlated with spondylodiscitis-associated mortality [25,26]. As our results show that the presence of intervertebral gas was associated with both higher CCI scores and higher CRP levels, the possibility of a more severe course of disease needs to be kept in mind for these patients.

Some limitations of our study need to be discussed. Due to our study’s retrospective design, inherent limitations and biases are present. An unknown bias may have been caused by missing imaging or microbiological data. Furthermore, the presence of intervertebral gas was scored at a single time-point, which is why pre-existing degenerative intervertebral gas could not be excluded. As follow-up data was not available, we were not able to analyze the long-term consequences of this phenomenon or mortality rates. As certain patient subgroups were small, the statistical power of these analyses was limited. However, to our knowledge, this is still the largest study of intervertebral gas in patients presenting with spontaneous spondylodiscitis. 

## 5. Conclusions

In conclusion, our results challenge the current notion of the presence of intervertebral gas ruling out spinal infection. We show that in about one-fifth of patients presenting with spontaneous spondylodiscitis, intervertebral gas was detected. Thus, especially in the case of further signs of spinal infection, spondylodiscitis should be considered, particularly in elderly patients, even if intervertebral gas is present. 

## Figures and Tables

**Figure 1 diagnostics-12-01089-f001:**
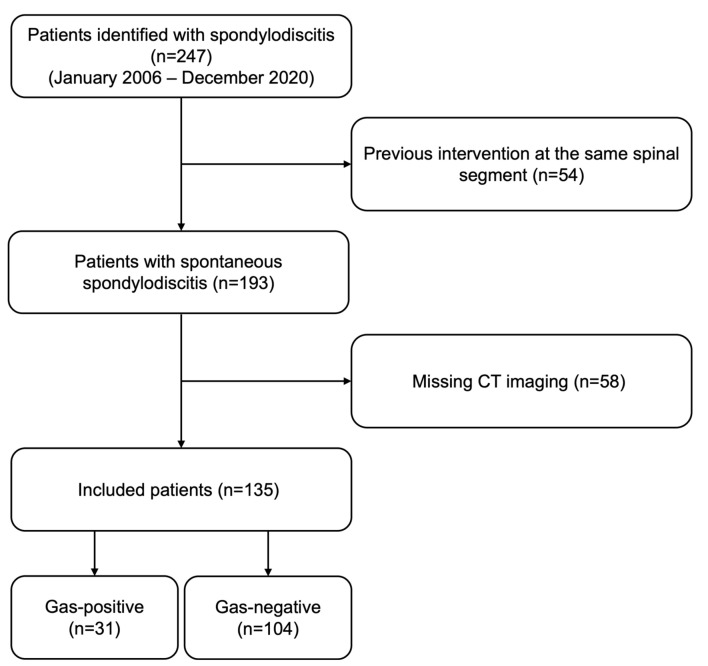
Flow chart of patient inclusion.

**Figure 2 diagnostics-12-01089-f002:**
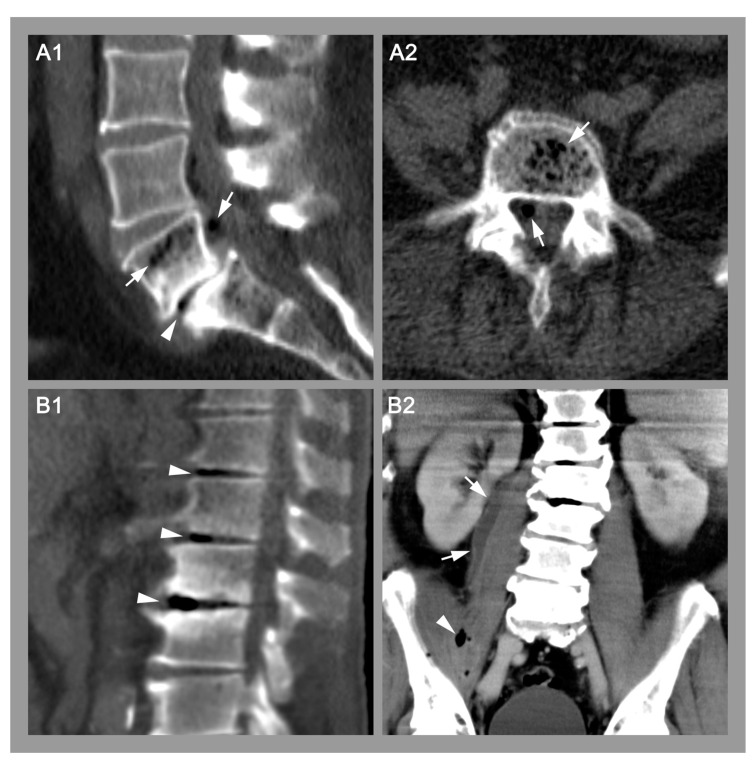
Imaging examples of two patients with confirmed spondylodiscitis. (**A**). Sagittal (**A1**) and transverse (**A2**) CT images showing gas within the disc (arrowhead), within the vertebra and within an epidural abscess (arrows) of a 50-year-old patient with urosepsis and spondylodiscitis L5/S1. Blood cultures and intraoperative tissue samples revealed *Escherichia coli* as the causative pathogen. (**B**). Sagittal (**B1**) and coronal (**B2**) CT images of intervertebral gas (**B1**, arrowheads) within three discs and gas (**B2**, arrowheads) within a psoas abscess (**B2**, arrows) of a 75-year-old patient with spondylodiscitis L1-3. Blood cultures and biopsy samples revealed *Streptococcus pyogenes* as the causative pathogen.

**Table 1 diagnostics-12-01089-t001:** Methods performed for pathogen identification in the two treatment groups.

	Conservative Treatment (*n* = 35)	Surgical Treatment (*n* = 100)
**CT-guided biopsy**	29 (82.9%)	18 (18.0%)
**Open biopsy**	2 (5.7%)	44 (44.0%)
**Intraoperative tissue sampling**	-	88 (88.0%)

**Table 2 diagnostics-12-01089-t002:** Patient characteristics of gas-positive and gas-negative patient groups. * indicates statistically significant results. BMI, body mass index; CCI, Charlson Comorbidity Index; IV, intravenous; HIV, human immunodeficiency virus; VAS, visual analog scale; CRP, C-reactive protein; WBC, white blood cell count; Hb, hemoglobin.

	Gas-Positive (*n* = 31)	Gas-Negative (*n* = 104)	*p*-Value
**Age**	71.9 ± 10.8	64.0 ± 13.9	0.004 *
**Gender (f:m)**	14:17	47:57	0.998
**BMI**	27.6 ± 6.1	26.4 ± 6.2	0.361
**CCI**	3.9 ± 2.9	2.7 ± 2.8	0.044 *
**Comorbidities**			
Smoking	7/31 (22.6%)	21/94 (22.3%)	0.978
Diabetes	9/31 (29.0%)	22/104 (21.2%)	0.360
Cancer	15/31 (48.4%)	28/104 (26.9%)	0.024 *
IV drugs	0/31 (0%)	1/104 (1.0%)	1.000
HIV	1/31 (3.2%)	1/104 (1.0%)	0.408
**Symptoms**			
Body temperature (°C)	37.2 ± 1.1	36.8 ± 0.7	0.044 *
Fever	6/29 (20.7%)	7/98 (7.1%)	0.035 *
Back pain at rest (VAS)	4.0 ± 2.6	3.9 ± 2.7	0.950
Back pain moving (VAS)	5.5 ± 2.7	5.7 ± 2.7	0.652
**Localization**			0.066
Cervical	1/31 (3.2%)	6/104 (5.8%)	
Thoracic	4/31 (12.9%)	33/104 (31.7%)	
Lumbar	25/31 (80.6%)	56/104 (53.8%)	
Disseminated	1/31 (3.2%)	9/104 (8.7%)	
**Lab values**			
CRP (mg/L)	134.2 ± 127.1	89.8 ± 97.3	0.040 *
WBC (/nL)	11.7 ± 6.2	10.2 ± 5.7	0.209
Hb (g/dL)	11.7 ± 2.0	11.3 ± 1.7	0.241
**Positive microbiology**	19/31 (61.3%)	74/104 (71.2%)	0.298
**Treatment**			0.986
Conservative	8/31 (25.8%)	27/104 (26.7%)	
Surgical	23/31 (74.2%)	77/104 (73.3%)	
**Duration of hospital stay (d)**	25.4 ± 17.6	22.1 ± 17.8	0.369

**Table 3 diagnostics-12-01089-t003:** Pathogen spectra found in gas-positive and gas-negative patient groups. * indicates statistically significant results.

	Gas-Positive (*n* = 19)	Gas-Negative (*n* = 74)	*p*-Value
** *Staphylococcus aureus* **	4/19 (21.1%)	24/74 (32.4%)	0.410
**Coagulase-negative staphylococci**	7/19 (36.8%)	22/74 (29.7%)	0.551
** *Propionibacterium acnes* **	2/19 (10.5%)	10/74 (13.5%)	0.729
**Streptococci**	3/19 (15.8%)	6/74 (8.1%)	0.382
**Enterococci**	3/19 (15.8%)	1/74 (1.4%)	0.026 *
** *Escherichia coli* **	3/19 (15.8%)	6/74 (8.1%)	0.382
**Other gram-positive**	0/19 (0%)	3/74 (4.1%)	0.372
**Other gram-negative**	3/19 (15.8%)	6/74 (8.1%)	0.382

## Data Availability

The datasets generated during and/or analyzed during the current study are not publicly available due to patient privacy but are available from the corresponding author on reasonable request.

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
