# Peer review of "Gas within the Intervertebral Disc Does Not Rule Out Spinal Infection—A Case Series of 135 Patients with Spontaneous Spondylodiscitis"

_diagnostics, 2022, doi:10.3390/diagnostics12051089_

Round 1

Reviewer 1 Report

In the presented manuscript, Schömig et.al reported that up to 23% of patients with spontaneous spondylodiscitis presented with intervertebral gas formation / or preexisting gas formation due to pior degenerative processes. The authors postulate that gas formation should not be used to rule out spondylodiscitis due to gas formation visible in CT images but must be interpreted in the context of further imaging and clinical findings, especially in elderly patients prone to preexisting degenerative disc disorders.

Overall, this study is interesting and solid in content.

Here are some minor comments:

  1. As different types of tissue sampling for microbiologic diagnostic were mentioned, did the authors find inferior results (proof of causative pathogen) using CT-guided biopsy vs. open biopsy, as this seems to have become the preferred diagnostic option?
  2. The authors describe correlation of the vaccum phenomenon and age, CCI as well as cancer diagnosis. As the ‘gas positive’ patients also presented with higher body temperature and higher CRP-levels:
    1. was there a correlation regarding time to diagnosis as higher CRP could be interpreted in a context of an advanced stage of disease (which might be due to preexisting back pain / degeneration which masked early symptoms) ?
    2. was there a correlation regarding mortality in this cohort as an older patient cohort with higher CCI has already proven more prone to a rapid disease progression and higher mortality in the context of spondylodiscitis?

Author Response

Response to Reviewer 1

In the presented manuscript, Schömig et.al reported that up to 23% of patients with spontaneous spondylodiscitis presented with intervertebral gas formation / or preexisting gas formation due to pior degenerative processes. The authors postulate that gas formation should not be used to rule out spondylodiscitis due to gas formation visible in CT images but must be interpreted in the context of further imaging and clinical findings, especially in elderly patients prone to preexisting degenerative disc disorders.

Overall, this study is interesting and solid in content.

  • Authors’ response: First of all, we would like to thank the reviewer for taking his/her time to critically read and comment on our manuscript. We are convinced that implementing his/her suggestions substantially improved the quality of our work especially regarding the discussion of our results.

Here are some minor comments:

  1. As different types of tissue sampling for microbiologic diagnostic were mentioned, did the authors find inferior results (proof of causative pathogen) using CT-guided biopsy vs. open biopsy, as this seems to have become the preferred diagnostic option?
  • Authors’ response: We most certainly agree with the reviewer that comparing different diagnostic options is of high importance. In the studied cohort, only six patients had both CT-guided biopsy and open biopsy which is why we do not feel confident to make an evidence-based claim in favor of either method based on our results. Looking at these six patients, there was only one patient who had a positive open biopsy with negative CT-guided biopsy. Comparing both techniques with intraoperative sampling yielded a higher diagnostic accuracy of open biopsy compared with CT-guided biopsy. However, again in our opinion the number of included patients is not high enough to allow definitive conclusions to be drawn. As these results still add to the existing literature, we included them in the revised version of our manuscript but also discussed their limited value.

“Diagnostic accuracies of microbiological tissue sampling methods were analyzed by cross tabulation with intraoperative tissue sampling as the diagnostic standard.”   

“A total of 47 patients underwent CT-guided biopsy with 25 (53.2%) positive results. Open biopsy was performed in 45 patients with 31 (68.9%) positive results. Intraoperative tissue samples were analyzed in 88 patients with 57 (64.8%) positive results. Cross tabulation analysis with intraoperative tissue sampling as the diagnostic standard yielded a sensitivity of 0.92 and specificity of 0.77 for open biopsy and a sensitivity of 0.80 and specificity of 0.33 for CT-guided biopsy.”

“Here, we found open biopsy to be more accurate compared with CT-guided biopsy with intraoperative tissue sampling as the diagnostic standard. However, only 16 patients had both CT-guided biopsy and intraoperative tissue sampling and only 37 patients had both open biopsy and intraoperative tissue sampling which is why these results need to be interpretated cautiously.”

  1. The authors describe correlation of the vaccum phenomenon and age, CCI as well as cancer diagnosis. As the ‘gas positive’ patients also presented with higher body temperature and higher CRP-levels:
    1. was there a correlation regarding time to diagnosis as higher CRP could be interpreted in a context of an advanced stage of disease (which might be due to preexisting back pain / degeneration which masked early symptoms) ?
  • Authors’ response: Thank you very much for this valuable comment. We agree that elevated CRP levels may be interpreted as an advanced stage of disease. While all CRP values were obtained at the patients’ admission, due to unspecific symptoms of the disease, we are not able to point out the exact point of disease onset. As we most definitely agree that this is an important aspect, though, we did include this in our revised discussion:

“Our results show that patients with spondylodiscitis may not only show intervertebral gas but present with significantly higher initial body temperatures (37.2±1.1 vs. 36.8±0.7 °C, p=0.044) and significantly higher initial CRP-levels (134.2±127.1 vs. 89.8±97.3 mg/L, p=0.040) compared with the gas-negative group. Therefore, spondylodiscitis needs to be considered as a possible diagnosis especially in case of symptoms suggestive of spondylodiscitis in combination with intervertebral radiolucency in CT imaging. Both elevated body temperatures and CRP-levels may also be indicative of a more advanced disease stage in patients with preexisting degenerative changes masking initial symptoms of spondylodiscitis.”

    1. was there a correlation regarding mortality in this cohort as an older patient cohort with higher CCI has already proven more prone to a rapid disease progression and higher mortality in the context of spondylodiscitis?
  • Authors’ response: We definitely agree that age and morbidity are important factors contributing to mortality in the context of spondylodiscitis. As in a substantial number of cases, treatment of included patients was initialized in our hospital but carried on in secondary hospitals, we are not able to analyze mortality rates with the necessary certainty. However, we thank the reviewer for this valuable remark and for pointing out the existing literature in this regard. In addition to including the suggested literature, we adapted our discussion accordingly:

"This work does not have the claim to distinguish between gas collections primarily caused by degenerative disc disease and gas collections caused by the infection itself. However, as we show intervertebral gas occurring in up to 23.0% of patients presenting with spontaneous spondylodiscitis, this phenomenon is not suitable to rule out spinal infection as a possible diagnosis. This is especially important as clinical signs of spondylodiscitis are unspecific and diagnosis therefore is often delayed by up to several months [16,17]. Furthermore, previous studies have shown both higher CCI scores and high CRP-levels at diagnosis to be significantly correlated with spondylodiscitis-associated mortality [25,26]. As our results show that the presence of intervertebral gas was associated with both higher CCI scores and higher CRP-levels, the possibility of a more severe course of disease needs to be kept in mind in these patients.”

As we most definitely agree that the missing analysis of mortality is a major limitation of our study, we put more emphasis on this aspect in our discussion’s limitations paragraph:

“Some limitations of our study need to be discussed. Due to our study’s retrospective design, inherent limitations and bias are present. Unknown bias may have been caused by missing imaging or microbiological data. Furthermore, the presence intervertebral gas was scored at a single time-point, which is why pre-existing degenerative intervertebral gas could not be excluded. As follow-up data was not available, we were not able to analyze long-term consequences of this phenomenon or mortality rates.”

Reviewer 2 Report

Many thanks to the authors for having presented their systematic review. The topic is interesting and has a potential interest for the orthopedic community. The concept of the study is well described, even if there are some aspects that are not immediately clear for readers. 

An important issue is that about 70% of the patients presented a proven spondylodiscitis. Moreover, how many patients underwent biopsy before surgery? Did the conservatively treated patients have a proven spondylodiscitis?

These aspects must be clarified.

Author Response

Response to Reviewer 2

Many thanks to the authors for having presented their systematic review. The topic is interesting and has a potential interest for the orthopedic community. The concept of the study is well described, even if there are some aspects that are not immediately clear for readers. 

  • Authors’ response: First of all, we would like to thank the reviewer for critically reading and commenting on our study and for his/her clear and precise suggestions which in our opinion helped considerably raise the quality of our work. Most importantly, we are convinced that this review process helped us significantly improve our results section which is highly important for a better understanding of our discussion and for the conclusions we draw from our analysis.

An important issue is that about 70% of the patients presented a proven spondylodiscitis. Moreover, how many patients underwent biopsy before surgery? Did the conservatively treated patients have a proven spondylodiscitis? These aspects must be clarified.

  • Authors’ response: Thank you very much for this comment. We definitely agree that a more precise depiction of our results substantially improves our manuscript and is absolutely necessary to convey our key messages. We therefore adapted our results section according to your and the second reviewer’s suggestions:

“Sixty-one patients (45.2%) were females. Mean patient age was 65.9±13.6 years, mean BMI 26.7±6.1 kg/m2. Seven patients (5.2%) had spondylodiscitis of the cervical spine, 37 patients (27.4%) of the thoracic, and 81 patients (60.0%) of the lumbar spine. In ten cases (7.4%), spondylodiscitis was disseminated. In 93 patients (68.9%) a causative pathogen was found. Thirty-five patients (25.9%) were treated conservatively, in 17 (48.6%) of which a causative pathogen was found by CT-guided or open biopsy. One hundred patients (74.1%) received surgical treatment. In 76 (76.0%) of these patients a causative pathogen was found. Methods performed for pathogen identification are shown in Table 1.

Table 1. Methods performed for pathogen identification in the two treatment groups.

Conservative treatment (n=35)

Surgical treatment (n=100)

CT-guided biopsy

29 (82.9%)

18 (18.0%)

Open biopsy

2 (5.7%)

44 (44.0%)

Intraoperative tissue sampling

-

88 (88.0%)

“A total of 47 patients underwent CT-guided biopsy with 25 (53.2%) positive results. Open biopsy was performed in 45 patients with 31 (68.9%) positive results. Intraoperative tissue samples were analyzed in 88 patients with 57 (64.8%) positive results. Cross tabulation analysis with intraoperative tissue sampling as the diagnostic standard yielded a sensitivity of 0.92 and specificity of 0.77 for open biopsy and a sensitivity of 0.80 and specificity of 0.33 for CT-guided biopsy.”

Round 2

Reviewer 2 Report

Persist to be not clear how many patients had a proven spondylodiscitis.

Tab. 3: check the table. I don't understant what the numbers means to. The sum is different from the total number of patients per group.

Author Response

Response to Reviewer 2:

Persist to be not clear how many patients had a proven spondylodiscitis.

  • Authors’ response: Thank you very much for again carefully reading and commenting on our manuscript. We highly appreciate your time especially since we are convinced that this process helps in significantly improving our work. We agree that a clear presentation of our study’s results is crucial and apologize if in the previous revision we were not able to fully address your comment. In our analysis, we defined spondylodiscitis according to the Infectious Diseases Society of America guidelines as a combination of radiological changes and clinical findings and only included patients who according to these criteria had signs of spondylodiscitis including elevated infectious laboratory parameters, back pain, and/or abscess formation in MRI. In 93 of 135 patients (68.9%) additionally a causative pathogen was found by CT-guided biopsy, open biopsy, or intraoperative tissue sampling, and thus these patients had a proven spondylodiscitis. Nineteen of these patients were gas-positive while 74 patients were gas-negative. We adapted our results section accordingly and hope that the presentation of our results is now more precise:

“Sixty-one patients (45.2%) were females. Mean patient age was 65.9±13.6 years, mean BMI 26.7±6.1 kg/m2. Seven patients (5.2%) had spondylodiscitis of the cervical spine, 37 patients (27.4%) of the thoracic, and 81 patients (60.0%) of the lumbar spine. In ten cases (7.4%), spondylodiscitis was disseminated. In all included patients, diagnosis of spondylodiscitis was based on a combination of radiological changes of the intervertebral disc and adjacent vertebrae as well as possible abscess formation in MRI and/or CT scans and clinical findings such as elevated CRP or WBC levels, back or neck pain, and fever. In 93 patients (68.9%) a causative pathogen was found.”

“Pathogen spectra are shown in Table 3. In 19 (61.3%) gas-positive patients and 74 (71.2%) gas-negative patients at least one pathogen was detected.”

Tab. 3: check the table. I don't understant what the numbers means to. The sum is different from the total number of patients per group.

  • Authors’ response: Again, thank you for taking the time to critically comment on our manuscript. In Table 3, we present the frequencies of pathogens found in the gas-positive and gas-negative groups. As in six gas-positive and four gas-negative patients multiple pathogens were found, the number of pathogens is higher than the number of patients. We apologize if this does not become clear from the manuscript and adapted it accordingly:

Pathogen spectra are shown in Table 3. In 19 (61.3%) gas-positive patients and 74 (71.2%) gas-negative patients at least one pathogen was detected. Polymicrobial infections were found in six (31.6%) gas-positive and four (5.4%) gas-negative patients.“